# The Utility of Metabolomics in Spinal Cord Injury: Opportunities for Biomarker Discovery and Neuroprotection

**DOI:** 10.3390/ijms26146864

**Published:** 2025-07-17

**Authors:** Prince Last Mudenda Zilundu, Anesuishe Blessings Gatsi, Tapiwa Chapupu, Lihua Zhou

**Affiliations:** 1Department of Basic Medical Sciences, College of Medicine, Ajman University, Ajman 346, United Arab Emirates; 2Department of Anatomy, University of Zimbabwe, Harare MP167, Zimbabwe; 3Department of Anatomy, Sun Yat-sen School of Medicine, Sun Yat-sen University, Shenzhen 518107, China

**Keywords:** spinal cord injury, neuroprotection, biomarkers, metabolomics

## Abstract

Brachial plexus root avulsion [BPRA] and concomitant spinal cord injury [SCI] represent devastating injuries that come with limited hope for recovery owing to the adult spinal cord’s loss of intrinsic ability to spontaneously regenerate. BPRA/SCI is an enormous public health issue the world over, and its catastrophic impact goes beyond the patient, the family, businesses, and national health budgets, draining billions of dollars annually. The rising population and economic growth have seen the incidence of SCI surging. Genomic, transcriptomic, and proteomic studies have yielded loads of information on the various molecular events that precede, regulate, and support both regenerative and degenerative pathways post-SCI. Metabolomics, on the other hand, comes in as the search for a cure and the objective monitoring of SCI severity and prognosis remains on the horizon. Despite the large number of review articles on metabolomics and its application fields such as in cancer and diabetes research, there is no comprehensive review on metabolite profiling to study disease mechanisms, biomarkers, or neuroprotection in SCI. First, we present a short review on BPRA/SCI. Second, we discuss potential benefits of metabolomics as applied in BPRA/SCI cases. Next, a look at the analytical techniques that are used in metabolomics. Next, we present an overview of the studies that have used metabolomics to reveal SCI metabolic fingerprints and point out areas of further investigation. Finally, we discuss future research directions.

## 1. Introduction

### 1.1. What Is Brachial Plexus Root Avulsion and Its Link with Spinal Cord Injury?

Brachial plexus root avulsion (BPRA) is a serious nerve injury accompanied by a concomitant reactive injury in the central nervous system, especially in the motor and sensory neurons of the spinal cord [1]. Avulsion of the brachial plexus roots leads to their detachment from the spinal cord, initiating a cascade of central pathological responses characterized by astrocytic and microglial activation, along with the degeneration and death of motor neurons whose axons have been severed [2,3]. It leads to the loss of sensation and motor function in the ipsilateral upper limb and often leaves the patients with chronic pain [4]. Numerous therapeutic strategies have been designed to address brachial plexus avulsion and concomitant spinal cord injuries in both humans and experimental animals [5]. These include surgery [6,7,8], cell transplantation [9], neurotrophin delivery [10], the removal of growth inhibition [11], immune modulation, and even the use of scaffolds for axonal guidance [12,13]. Despite all these attempts, there is still no cure or clinically effective therapy currently available for BPRA or spinal cord injuries [14,15]. There is now a consensus that, due to the complexity of the injured tissue environment, a multi-faceted treatment regime is necessary [16], and that the integration of multiple technologies is likely key to the effective treatment of spinal cord injuries. Accordingly, further extensive preclinical studies are indicated. Among the technologies poised to transform our understanding of the complex pathological environment of BPRA and SCI is the integration of metabolomics into both preclinical and clinical studies. However, there is a dearth of studies on the application of metabolomics in BPRA [17,18].

### 1.2. Enter Metabolomics and Metabonomics

Metabolomics is an emerging field, similar to proteomics, transcriptomics, and genomics, that is specifically meant for high-throughput global profiling of the collection of all low-molecular-weight (<1 kDa) metabolites in a biological system (i.e., the metabolome) [19,20]. These include amino acids, fatty acids, sugars, lipids, and steroids [21,22]. It provides a snapshot of the metabolic dynamics that reflect the response of living systems to both pathophysiological stimuli and/or genetic modification [23,24]. On the other hand, metabonomics, a subcategory of metabolomics, focuses on the temporal interactions between metabolites over time [24]. Each type of cell and tissue has a characteristic metabolic composition that is uniquely altered in response to physiological and pathophysiological stimuli, making knowledge of such dynamic compositions of great clinical and scientific interest [20]. These characteristic metabolic compositions are a result of the combined influence of epigenetic factors, heterogeneous distributions of molecules, and differential reaction rates [24]. Metabolomics, therefore, involves the examination of biofluids, including blood plasma/serum, urine, feces, blister fluid, saliva, and semen, as well as tissue extracts and intact tissue biopsies to study many diseases [25,26,27,28]. For example, metabolomics has seen many applications in diseases such as cancer [29,30], diabetes [31,32], tuberculosis [33], degenerative central nervous system disease [34], and even drug discovery [35]. Metabolomics is also currently being used to discover new biomarkers of various diseases and to identify biochemical pathways involved in their pathogenesis [36].

### 1.3. What Will Metabolomics Do to BPRA/SCI?

A range of metabolic derangements has been observed following brachial plexus root avulsion (BPRA) and spinal cord injury (SCI), including oxidative and nitrosative stress, disrupted glycolysis, and altered amino acid and lipid metabolism. Notably, even when axonal injury is peripheral, as in BPRA, significant metabolic responses occur both locally and within the spinal cord, underscoring a shared pathophysiological continuum between BPRA and SCI [17,37]. The primary injury phase in BPRA involves axonal severance, neuronal death, and vascular disruption, while secondary processes in the spinal cord include edema, ion imbalance, and apoptotic and necrotic cell death [14,38]. These changes result in the accumulation of reactive oxygen species, neurotransmitters, and cytotoxic debris, which further fuel inflammation and neuronal loss [3]. Despite this degenerative cascade, limited spontaneous improvement sometimes occurs, driven by endogenous repair mechanisms such as spinal tissue remodeling [39].

Understanding these metabolic disturbances is critical for uncovering injury mechanisms, monitoring progression, and guiding treatment strategies. While transcriptomic and proteomic studies have revealed the involvement of molecules like c-Jun and nNOS in spinal motor neurons following BPRA [40,41], metabolomics remains underexplored in this context [18]. This gap is significant, as both BPRA and SCI induce metabolic signatures with strong implications for neuroprotection and regeneration. Given the limitations of current clinical scales such as the ASIA Impairment Scale, which can be imprecise and often infeasible to use in acute trauma settings, there is an urgent need for objective biomarkers to stratify injury severity and predict outcomes [15,42,43,44].

Metabolomics offers a promising solution. By capturing a real-time chemical fingerprint of the body’s physiological and pathological states, it provides higher diagnostic sensitivity than protein-based analyses [45]. It enables the identification of disease-specific metabolic signatures essential for precision medicine [46], helping to elucidate injury mechanisms, discover neuroprotective biomarkers, and uncover therapeutic targets. Metabolite profiles reflect genome–environment interactions [47], making metabolomics especially powerful in the study of complex injuries like SCI. As such, metabolomics stands to redefine how we classify, monitor, and treat BPRA and SCI, marking a critical shift toward systems-level, personalized approaches in neurotrauma research.

## 2. The Literature Search Strategy

A comprehensive search of the literature was conducted to identify relevant peer-reviewed studies focusing on the application of metabolomics in spinal cord injury (SCI) and brachial plexus root avulsion (BPRA). The electronic databases PubMed, Scopus, and Google scholar, and specific journal publishers such as Wiley, MDPI, and Web of Science were searched for articles published between January 2017 and March 2025. The search strategy combined keywords and MeSH terms including: “metabolomics,” “metabonomics,” “spinal cord injury,” “brachial plexus injury or roots avulsion,” “neuroprotection,” “metabolic profiles,” and “biomarker discovery.” Boolean operators [“AND,” “OR”] and filters, e.g., animal studies, human clinical trials, were applied to refine results.

### 2.1. Inclusion and Exclusion Criteria

Studies were included if they:

Utilized metabolomic techniques (e.g., LC-MS, GC-MS, and NMR) to analyze tissue, blood, urine, feces or cerebrospinal fluid (CSF) samples.

Investigated either animal models or human subjects with SCI or BPRA.

Reported outcomes related to metabolic changes, biomarker identification, or therapeutic targets.

### 2.2. Exclusion Criteria Were

Non-metabolomics-based studies.

Conference abstracts, editorials, and commentaries without primary data.

Non-English publications.

### 2.3. Study Selection Process

Two independent reviewers screened the titles and abstracts of the retrieved articles for relevance. The full texts of potentially eligible studies were then assessed for inclusion. Disagreements were resolved through discussion or third-party adjudication. Figure 1 shows the PRISMA study flow diagram [48].

### 2.4. Data Extraction and Synthesis

Data were systematically extracted using a pre-defined template, including the following:

Author(s), year, and country of study; animal or human model used; tissue or biofluid analyzed; analytical technique employed (e.g., LC-MS, NMR); key identified metabolites; and clinical or experimental outcome relevance (e.g., injury severity markers, neuroprotection indicators). Quantitative synthesis was limited due to methodological heterogeneity; therefore, results were summarized descriptively. Table 1 and Table 2 were used to organize studies by species, sample source, technique, and biomarker application.

## 3. Analytical Techniques Used in Metabolomics

Metabolites comprise a broad array of low-molecular-weight structures, such as lipids, amino acids, and carbohydrates, making their comprehensive analysis challenging. With the advancement in analytical technologies like Gas Chromatography, Ultra Performance Liquid Chromatography (UPLC), and Nuclear Magnetic Resonance (NMR), it is now feasible to identify, quantify, and study these metabolites and their metabolic pathways. These metabolomics technologies have been described in detail recently [90,91]. However, due to the vast diversity and complexity of the metabolome, no single technology can detect all metabolites in a given biological sample. This section delves into the latest methods used in metabolomics for metabolic profiling.

The most common analytical platforms used in metabolomics involve nuclear magnetic resonance (NMR) spectroscopy, mostly 1H-NMR, and mass spectrometry [MS]. Both permit the analysis of a multitude of small molecules coexisting in a sample including metabolite identification and quantitation, but each of them has its own strengths and limitations. Huang et al. [92] recently provided a succinct review of NMR-based metabolomics practices in human disease research. Briefly, NMR spectroscopy offers benefits such as minimal sample preparation and the ability to analyze without separating analytes. Its non-destructive approach ensures that specimens remain intact for other analyses, making it ideal for studying intact cells or tissue samples such as nervous tissue. Furthermore, NMR signal intensity is less impacted by sample matrix components compared to the MS technique, making it well-suited for complex biological samples like cerebrospinal fluid. However, NMR’s primary drawback is its low sensitivity. Numerous factors such as diet, stress, and age, can influence metabolic composition, posing challenges in biomarker discovery [45]. These elements are easier to control in lab animals compared to humans. For uniform outcomes and replicability, it is vital to optimize every experimental step, from gathering samples to analyzing data, in accordance with local standard guidelines [93].

Application of mass spectrometry in metabolomic investigation usually requires the separation of metabolites from analyzed biofluids. Thus, mass spectrometry is frequently coupled with high-resolution techniques, such as liquid chromatography [LC], gas chromatography (GC), or capillary electrophoresis (CE) [94]. Chromatography–mass spectrometry platforms have the advantage of higher sensitivity, hence their use over NMR. Although mass spectrometry is a destructive technique, it has the advantage of only requiring minute amounts (μL) of the sample for analysis. In GC-MS, substances are heated until they become gaseous, so non-volatile materials cannot be directly studied and need derivatization. With the right choice of mobile phases and column, LC-MS allows for the examination of an extensive array of metabolites, from hydrophilic to hydrophobic [45,95]. Upgrading of chromatography has seen the introduction of ultra-high-performance liquid chromatography (UHPLC). The UHPLC systems use columns with smaller particle sizes, which result in increased peak capacity, higher sensitivity, and the reduced time of analysis in comparison to HPLC [96]. The separation mechanism in capillary electrophoresis is different from that used in liquid and gas chromatography and enables analysis of polar and ionic molecules. Low repeatability is one of the major limitations of CE, whereas excellent separation capacity is regarded as the main advantage of this analytical technique [97].

## 4. Attempts to Get Down to Metabolomics in SCI

In recent years, the exploration of metabolomics has emerged as a promising frontier in understanding the intricate biochemical landscapes of spinal cord injuries. The scientific development of novel therapies for SCI treatment depends upon an understanding of the complex pathophysiologic mechanisms that are triggered after acute SCI. These underlying processes have been studied extensively at the level of proteomics, transcriptomics, and genomics (-omics). However, the contributions of metabolomics have not been on par in spite of the work in progress as outlined below [98].

### 4.1. SCI Animal Studies Involving Metabolomics

A growing body of metabolomics research in preclinical spinal cord injury animal models has elucidated a complex but increasingly coherent portrait of injury-induced biochemical disturbances. As summarized in Table 1, these studies employ a range of high-resolution analytical techniques—including untargeted and targeted liquid chromatography–mass spectrometry (LC-MS/MS), gas chromatography–mass spectrometry (GC-MS), and mass spectrometry imaging (MSI)—to interrogate the metabolic landscape of SCI across rat and mouse animal models, injury mechanisms, and tissue types.

The metabolomic analyses in Table 1 were typically conducted on *n* = 3 to *n* = 8 animals per group, across tissues like spinal cord, plasma, serum, CSF, urine, feces, skeletal muscle, lung, and exosomes, with time points ranging from 30 min to 28 days post-injury, reflecting a broad representation of SCI pathophysiology in preclinical animal models. A central and recurring theme is the dysregulation of lipid metabolism following SCI. Studies utilizing both targeted and untargeted LC-MS/MS approaches have consistently identified arachidonic acid derivatives (such as PGE2 and LTB4 [49,67]), lysophospholipids [76], and a spectrum of fatty acids and their derivatives [50,55,66,68,76] as robust biomarkers of acute neuroinflammation, membrane integrity, mitochondrial dysfunction, and secondary injury progression. For example, a study by Yang et al. [61] demonstrated that uric acid, phosphorylcholine, pyridoxine, and guanidoacetic acid, differentially altered across spinal cord, CSF, and plasma, serve as candidate biomarkers for spinal cord injury severity, reflecting disruptions in glyoxylate, dicarboxylate, glycine, serine, and threonine metabolism linked to neuroinflammation, oxidative stress, and membrane repair.

In addition, Liang et al. [55] further demonstrated that glycerophospholipid metabolites such as phosphatidylserines, phosphatidylethanolamines, and phosphatidylcholines were identified as biomarkers of neuropathic pain in rats, reflecting disrupted membrane homeostasis and neuroinflammation mediated via microglial P2 × 4/IRF8/P38 MAPK signaling and oxidative stress pathways. The prominence of these lipid biomarkers likely reflects the universal activation of phospholipases and lipid peroxidation signaling pathways in response to membrane disruption and oxidative stress [55]. These processes are rapidly engaged regardless of the specific injury mechanism and persist despite distinct metabolic profiles. These findings collectively underscore that lipidomic dysregulation post-SCI centers around phospholipid degradation, fatty acid oxidation, and inflammation. Lipid species such as lysophospholipids, oxylipins, and carnitines may serve as robust biomarkers or therapeutic targets in SCI pathology and recovery monitoring.

In parallel, multiple studies in Table 1 report significant disturbances in amino acid metabolism following SCI, with key neurotransmitter precursors and derivatives, including dopamine, tyrosine [51,68], tryptophan [56,60], glutamate [62,64,65,70], alanine [62,80], and branched-chain amino acids [73], emerging as functional biomarkers of neuropathic pain, neuroinflammation, and metabolic dysfunction [59,73]. Although the specific metabolite profiles vary between studies, dopamine, tyrosine, tryptophan, and their downstream derivatives have been consistently reported as altered following SCI, particularly in experimental models of neuropathic pain and opioid responsiveness [51,56,68]. Additionally, Liu et al. [59] demonstrated that metabolite alterations in the peripheral detrusor muscle, particularly in acetylcholine, histamine, and kynurenine pathways, reflect neuromodulatory and neuroimmune disruption beyond spinal cord or blood biomarkers, supporting their potential role in identifying neurogenic bladder dysfunction following SCI. These metabolites are not only implicated in pain modulation and central sensitization but are also linked to gut–brain axis signaling, with Kang et al., 2023, and Chen et al., 2021, independently demonstrating that alterations in amino acid metabolism are influenced by changes in the gut microbiota [68,73]. Disruption of the tyrosine metabolism is a shared feature in Rodgers et al. [51] and Yuan et al. [56], though the specific metabolites differ. Rodgers and colleagues linked altered dopamine-related metabolites to morphine resistance, identifying markers of opioid responsiveness. Yuan et al. found that ligustrazine and sinomenine modulated tyrosine pathway balance, highlighting different tyrosine-derived metabolites as therapeutic indicators [56]. Despite variation, both studies support tyrosine metabolism as a core disrupted pathway and a promising source of biomarkers for pain severity, neuroinflammation, and therapeutic efficacy.

Energy metabolism is another axis of convergence. SCI-induced energy metabolism disruption is consistently demonstrated across multiple tissues, with Liu et al. [59] identifying phosphocreatine, α-ketoglutaric acid, and fructose 1,6-bisphosphate in bladder muscle [54]; Zeng et al. [58] reported purine cycle markers [AMP, IMP, GMP, and guanidoacetic acid] in the spinal cord; Zhang et al. [54] detected pyruvate, lactate, and isocitrate in the serum; Shi et al. [72] confirmed lactate and glycolytic intermediates [glucose-6-phosphate, fructose-6-phosphate, and AMP] in the spinal tissue; and Graham et al. [72,79] and Potter et al. [78] revealed glucose, lactate, and pyruvate dysregulation in the skeletal muscle, all supporting impaired ATP production as a hallmark of SCI pathology and recovery potential. Despite this convergence in key metabolic pathways, the specific lists of altered metabolites often vary across studies due to differences in animal species [rats vs. mice], injury models [contusion, compression, or avulsion], sampling time points, and analytical conditions. These metabolites serve as sensitive markers of ATP depletion, mitochondrial dysfunction, and oxidative stress, and their detection is facilitated by the high sensitivity of both LC-MS/MS and GC-MS platforms. These metabolites not only reflect underlying mitochondrial and glycolytic dysfunction but also serve as candidate biomarkers for early diagnosis, severity stratification, and the evaluation of metabolic-targeted therapies in spinal cord injury, regardless of the model or tissue examined.

Despite these overarching similarities, important differences emerge that are closely tied to the mechanism of injury, the tissue or biofluid analyzed, and the analytical platform employed. Contusion models simulating blunt trauma emphasize acute inflammatory lipids and energy metabolites, such as prostaglandin E2 and leukotriene B4 [49] and purines and glycolytic intermediates like AMP and lactate [54,72]. In contrast, compression and CCI models highlight chronic osmotic and membrane stress, with elevated osmolytes like betaine and creatine [52] and metabolites such as taurine, ascorbic acid, and dopamine [58,68]. Transection and excitotoxic models reveal ionic imbalance and edema, marked by small osmolytes like carnosine and inosine [56,66]. By revealing how injury-type-specific metabolic profiles shape biomarker expression, this comparison supports a future stratified approach to biomarker discovery in SCI, ensuring relevance to both acute and chronic clinical scenarios.

The spatial resolution afforded by mass spectrometry imaging and region-specific sampling strategies has revealed that certain metabolites are localized to distinct pathological zones following SCI. For example, phosphatidylserine (PS) and related lipids were identified in inflamed spinal cord regions using AFADESI-MSI, marking sites of oxidative stress and neural damage [66], while nicotinamide was enriched in fibrotic scar regions and proposed as a marker of scar modulation and therapeutic efficacy [74]. Similarly, Pukale et al. [52] used MSI to detect osmolyte gradients, such as betaine and creatine, along the rostrocaudal axis of syrinx expansion. These findings underscore the importance of accounting for both anatomical location and injury phase when interpreting metabolomic profiles and selecting spatially relevant biomarkers.

The clinical and translational implications of these findings are substantial. Many of the identified metabolites are not only diagnostic or prognostic indicators, reflecting injury severity, progression, or comorbid risks such as neuropathic pain and depression, but they also point toward therapeutic potential. For instance, docosahexaenoic acid (DHA) was proposed by Jiang et al. [57] and Zhang et al. [60] as a neuroprotective and anti-inflammatory biomarker linked to repair and resolution signaling across species and platforms. Similarly, different types of short-chain fatty acids (SCFAs), including butyrate and propionate, were associated with the gut–brain axis recovery and anti-inflammatory activity [63,68,71,75]. Moreover, exosomal fatty acids linked to depression-like behavior post-SCI [63] and nicotinamide (NAM), implicated in fibrotic scar suppression [74], are highlighted as mechanistic biomarkers with therapeutic relevance. Integrating metabolomics with transcriptomics, proteomics, and spatial mapping strategies is likely to refine these biomarker panels, improving their specificity, contextual relevance, and utility in both experimental and clinical settings.

In summary, rat and mouse SCI models contribute distinct but complementary insights into biomarker discovery. Rat studies (e.g., [54,58]) primarily highlight acute-phase markers of inflammation, energy failure, and osmotic imbalance. In contrast, mouse studies (e.g., [57,72,73]) emphasize chronic metabolic remodeling, mitochondrial dysfunction, and gut–brain axis disruption. Together, these models support time-specific and system-wide biomarker identification, enhancing translational relevance in SCI. In conclusion, the collective evidence from Table 1 demonstrates that while certain metabolic disturbances, particularly in lipid, energy, and amino acid metabolism, are conserved across SCI models, the precise biomarker signatures are shaped by the mechanism of injury, tissue context, and analytical technique. Future studies should prioritize cross-model integration, longitudinal sampling, and validation in human cohorts to establish robust, phase-specific biomarker panels for diagnosis, prognosis, and therapeutic monitoring.

### 4.2. Human Studies Involving Metabolomics

The human SCI metabolomics studies listed in Table 2 consistently used advanced MS or NMR platforms on urine, serum, or plasma, and identify amino acids, energy metabolites, lipids, and gut-derived molecules as robust, non-invasive biomarkers. These metabolites track with injury severity, recovery, and intervention response, supporting their clinical utility for diagnosis, prognosis, and therapy monitoring in SCI.

The studies summarized in Table 2 collectively highlight the rapid progress and diversity of metabolomics approaches applied to biomarker discovery in human spinal cord injury patients. Across these investigations, a range of analytical techniques, including ^1^H NMR spectroscopy, LC-MS/MS, UHPLC-QTOF/MS, and GC-MS, were employed to interrogate metabolic changes in accessible biofluids such as urine, serum, plasma, and, in some cases, feces. The choice of technique and sample type not only influenced the breadth and specificity of metabolites detected but also shaped the clinical relevance and translational potential of the identified biomarkers. However, despite these advances, the routine clinical translation of these metabolomic biomarkers remains limited due to challenges in validation, standardization, and integration into existing diagnostic frameworks.

A consistent finding across clinical metabolomics studies is the disruption of amino acid metabolism following spinal cord injury, with several studies identifying amino acid-related metabolites as sensitive indicators of injury status and recovery. In serum samples, Singh et al. [82] reported significant alterations in glycine, valine, phenylalanine, alanine, tyrosine, isoleucine, and others, with glycine and lactate inversely correlating with neurological recovery scores. In a related longitudinal urine-based study, Singh et al. [84] again observed the significant regulation of alanine and phenylalanine, particularly in patients undergoing stem cell therapy, supporting their role as non-invasive biomarkers for monitoring functional improvement, as they were correlated with ASIA Impairment Scale scores. These consistent findings across time points and sample types highlight the translational potential of amino acid metabolites as accessible, reproducible indicators of neurological status, offering a path toward personalized monitoring tools in the clinical management of SCI patients. However, broader validation and standardization are still needed to establish their routine utility in clinical settings. Bykowski et al. [85] further identified L-valine in urine as a robust prognostic biomarker, highlighting its recurrence across studies; although not identical in context, its repeated detection underscores the relevance of branched-chain amino acids as a metabolite class in SCI biomarker discovery. Meanwhile, Li et al. [86] reported elevated phenylacetylglutamine and L-5-oxoproline in serum samples collected from chronic SCI patients, associating these metabolites with metabolic and cardiovascular risk, while Zhou et al. [77] detected the amino acid derivative S-allyl-L-cysteine in plasma following a rehabilitation program, suggesting links to neuroprotective processes. The authors of both studies integrated KEGG pathway analysis to interpret the biological relevance of their metabolomics findings. In Li et al. [86], phenylacetylglutamine and L-5-oxoproline, identified as elevated in SCI patients with prediabetes/type 2 diabetes, were linked to microbial metabolism and glutathione metabolism, respectively. These metabolites are implicated in cardiovascular and metabolic risk, reflecting host–microbe co-metabolism and oxidative stress responses. In contrast, Zhou et al. [77] reported S-allyl-L-cysteine following rehabilitation, mapping it to cysteine and methionine metabolism. This pathway supports neuroprotective mechanisms, including antioxidant defense and cellular repair. Despite variability in exact metabolite identities, the KEGG-mapped pathways converge on conserved mechanisms like oxidative balance, amino acid metabolism, and gut microbial contributions, reinforcing their pathophysiological relevance in SCI. These findings underscore the potential of targeting conserved metabolic pathways in biomarker discovery and highlight the need for longitudinal studies to validate their diagnostic and prognostic value in clinical settings.

The observed variations in metabolite profiles can be partly attributed to differences in biofluid type (serum vs. urine vs. plasma), time post-injury (acute vs. chronic), and therapeutic intervention status (e.g., stem cell therapy, exercise rehabilitation). Urinary and serum studies also differ in sensitivity and metabolite composition due to renal filtration and reabsorption dynamics. Additional clinical factors such as injury severity (e.g., AIS grade), presence of infections, particularly urinary tract infections common in SCI patients, medication use, comorbidities like diabetes or pressure ulcers, and nutritional status further complicate metabolic readouts, underscoring the need for careful interpretation and standardization in human SCI metabolomics research. Notably, L-valine and phenylalanine are the only amino acids consistently reported across both serum and urine studies [82,84,85], reinforcing their translational value as stable, non-invasive biomarkers for SCI monitoring.

Disruption of energy metabolism is a recurring and mechanistically relevant finding across multiple clinical SCI metabolomics studies. Serum-based analyses by Singh et al. [82] identified significant alterations in lactate, acetate, succinate, and glucose, all indicative of acute glycolytic flux and mitochondrial stress. These changes correlated with neurological recovery, especially as lactate and glycine levels decreased over time. In urine, Singh et al. [84] reported changes in acetate and β-hydroxybutyrate, a ketone body, alongside creatine and creatine phosphate, reflecting systemic metabolic adaptation during subacute recovery. Bykowski et al. [87] confirmed the serum elevation of lactate, acetic acid, and succinic acid up to six months post-injury, linking them to functional recovery (SCIM score). Further, Shi et al. [71] measured elevated lactate, glucose-6-phosphate, and fructose-6-phosphate in spinal cord tissue, emphasizing local neuronal energy failure and impaired axonal regeneration. Although not focused on traditional energy intermediates, Li et al. [86] also identified glutamine and L-5-oxoproline, linked to energy and redox balance, in chronic SCI serum samples, particularly in patients with metabolic comorbidities (e.g., prediabetes).

Lactate and acetate emerged as the most consistently altered energy metabolites across studies and sample types (e.g., [84,85,88]), suggesting their robustness as systemic biomarkers. However, differences in reported metabolites likely reflect variations in sample type (serum vs. urine vs. tissue), analytical platform (NMR vs. MS), injury chronicity, and cohort characteristics. For example, serum and tissue samples captured acute mitochondrial and glycolytic disturbances, while urine-based studies (e.g., Singh et al. [84]) may emphasize renal clearance-filtered metabolic byproducts. Chronic studies, e.g., Li et al. [86] further showed energy metabolites associated with oxidative stress and cardiometabolic risk, reflecting longer-term adaptation or dysfunction. Collectively, these findings highlight a core panel of energy-related metabolites, particularly lactate, acetate, succinate, and β-hydroxybutyrate, as promising biomarkers for metabolic dysfunction, therapeutic response, and longitudinal recovery monitoring in SCI.

Four studies in Table 2 highlighted lipid metabolism as a promising domain for biomarker discovery in SCI, particularly in relation to neuroinflammation, immune modulation, and rehabilitation outcomes. Using untargeted LC-MS/MS, Zhou et al. [77] identified elevated lysoglycerophospholipids (e.g., LPE[P-17:0]) and free fatty acids in plasma following a 4-week exercise program, with several correlating with immunoglobulin heavy-chain expression and functional recovery. Similarly, Kong et al. [88] reported increased levels of phosphatidylcholine PC [18:2/0:0] in serum from chronic SCI patients, suggesting persistent lipid remodeling linked to gut dysbiosis and systemic metabolic stress. Zhang et al. [89] extended these findings by detecting elevated fecal fatty acids, such as linoleic and butyric acid, implicating gut-derived lipid alterations in inflammation and recovery. Adding to this, Yarar-Fisher et al. [99] used untargeted metabolomics to show that a ketogenic diet increased serum levels of the anti-inflammatory lysophospholipid lysoPC [16:0] while decreasing pro-inflammatory fibrinogen peptides in acute SCI patients—changes that were associated with improved motor recovery. While these studies vary in biofluid type, patient chronicity, and intervention (e.g., diet vs. exercise), they converge on the dysregulation of lipid signaling, particularly phospholipids and fatty acids, as a core metabolic feature of SCI. Together, these findings support the translational potential of lipid-derived metabolites as mechanistic markers of neuroimmune regulation, metabolic recovery, and therapeutic response in spinal cord injury.

While Zhou et al. [77] focused on lipid changes associated with rehabilitation-induced neuroplasticity, Kong et al. [88] profiled lipid signatures at a later post-injury stage [~23 months], likely capturing chronic-phase metabolic adaptations. This difference in time post-injury, alongside distinct sample matrices (plasma vs. serum), may explain the variability in specific lipid species reported. Notably, both studies converge on the importance of glycerophospholipid metabolites, reinforcing their potential as common lipid biomarkers in SCI. Together, these studies support lipidomics, particularly the profiling of lysophospholipids and phosphatidylcholines, as a valuable approach for identifying biomarkers of chronic inflammation, repair, and rehabilitation response in human SCI.

Microbiota-derived metabolites and their interplay with systemic metabolism represent a rapidly emerging area, as exemplified by Li et al. [86], Kong et al. [88], and Jing et al. [81]. These studies combined metabolomics with microbiome profiling [using both MS and GC-MS for SCFAs] to show that metabolites such as indoxyl sulfate, phenylacetylglutamine, and short-chain fatty acids [butyric and valeric acid] are significantly altered in SCI patients, particularly those with metabolic comorbidities. The authors propose these metabolites as biomarkers for metabolic and cardiovascular risk as well as indicators of gut–brain axis disruption [71,81,86,89]. Notably, the reduction in beneficial SCFAs and the increase in gut-derived uremic toxins were linked to both injury severity and neurological outcome, highlighting the translational potential of these markers for risk stratification and therapeutic targeting [71,75,89].

Despite methodological differences, a consistent theme is the clinical utility of these biomarkers. Most studies emphasize the feasibility of using urine and serum for repeated, non-invasive sampling, thereby facilitating longitudinal monitoring in both acute and chronic SCI settings. The integration of metabolomics data with clinical outcome measures and, in some cases, machine learning approaches, has further refined the prognostic value of these markers. For instance, the combination of metabolite panels with SCIM scores [85,86] or with immunoregulatory protein levels [87] has yielded biomarker signatures with strong predictive power for neurological recovery and rehabilitation response.

While the human SCI metabolomics studies in Table 2 provide valuable insights, they are constrained by several methodological limitations that must be addressed to improve clinical utility and reproducibility. A primary concern is small sample sizes; for example, Bykowski et al. [85] included only six male patients, and their follow-up study [88] involved just seven male participants. Similarly, Singh et al. [82] and Singh et al. [100] enrolled approximately 20 SCI patients, limiting statistical power and increasing the risk of false-positive or false-negative findings. Zhou et al. [77] acknowledged that their small cohort reduced their ability to detect subtle multi-omic effects. Moreover, Bykowski et al. [87] highlighted substantial heterogeneity in injury level and severity within their cohort, reinforcing the need for larger, multi-center studies to enable subgroup analysis and validation.

Participant selection and control group design also varied significantly. Both Bykowski studies [85,87] enrolled only male participants, excluding females and potentially overlooking sex-specific metabolic signatures, an issue also noted by Zhou et al. [77] in relation to sex mismatches between human and animal subjects. Additionally, Singh et al. [82] focused on acute, complete injuries, whereas Kong et al. [88] included patients with a broad range of severities (AIS A–D) and chronicity (~23 months post-injury), introducing biological variability. Control group inclusion was also inconsistent: Kong et al. [88] and Jing et al. [81] included well-matched healthy controls, while Bykowski et al. [85,87] relied solely on within-subject longitudinal comparisons and lacked healthy or trauma-exposed controls, limiting their ability to distinguish SCI-specific effects from general post-trauma alterations.

Analytical platform limitations also influenced the findings. Several studies [82,84,85,87] used ^1^H-NMR, which provides high reproducibility but limited sensitivity, possibly missing low-abundance yet biologically important metabolites. In contrast, Kong et al. [88] applied high-resolution untargeted mass spectrometry (UHPLC-QTOF/MS), detecting over 5000 features but identifying only 41 named metabolites—highlighting the common challenge of poor annotation in untargeted MS studies. Future work could combine NMR and MS approaches to maximize coverage and follow up untargeted results with targeted validation using authentic standards.

Finally, study design and sample collection timing were suboptimal. Many studies used only two time points (e.g., baseline and 6 months in Singh et al. [84] and Bykowski et al. [85,87], missing intermediate dynamics between the acute, subacute, and chronic phases. Others, like Kong et al. [88] and Jing et al. [81], employed cross-sectional designs, limiting causal inference. Dietary intake, medications, and rehabilitation status were rarely controlled or recorded, although Bykowski et al. [85] acknowledged that caffeine levels were likely influenced by uncontrolled dietary factors. Moving forward, studies should adopt longitudinal sampling frameworks with more frequent time points, standardize collection conditions (e.g., fasting, morning sampling), and record potential confounders like diet, medication, and BMI.

### 4.3. Conserved Metabolic Pathways Across Animal and Human Studies

While animal studies offer mechanistic insights through controlled injury models and diverse tissue sampling, human SCI metabolomics reflects a broader spectrum of clinical variability, comorbidities, and interventions. Despite methodological differences, both domains converge on key pathways, energy metabolism, oxidative stress, amino acid dysregulation, and microbial metabolites, highlighting conserved biological responses to SCI and informing biomarker discovery across species.

Energy metabolism was the most consistently disrupted pathway. Shi et al. [71] showed that spinal cord injury-induced hypoxia suppresses the MCT1-mediated lactate transport, impairing neuronal energy supply and axonal regeneration in mice. Restoration of this shuttle improved recovery, highlighting lactate as a mechanistic and therapeutic target. In rats, studies by Ni et al. [53], Zhang et al. [54], and Lyu et al. [66] also highlighted elevated lactic acid, pyruvate, and inosine in the spinal cord and serum, indicating mitochondrial dysfunction and oxidative stress in acute and chronic phases. In human studies, Bykowski et al. [87] reported that lactate, acetic acid, and succinate levels in serum correlated with SCIM scores, while Singh et al. [84] found that urinary acetate and β-hydroxybutyrate tracked with functional recovery, demonstrating translational alignment in energy biomarkers across species.

Amino acid metabolism, especially branched-chain amino acids (BCAAs), also recurred as a conserved signature. In rats and mice, Kang et al. [73] and Zhang et al. [74] observed significant perturbations in valine, leucine, and isoleucine, correlating with neuroinflammation and gut–brain axis disruption. Human studies mirrored these findings: Singh et al. [82] reported that valine, isoleucine, and glycine levels were linked to neurological recovery, and Bykowski et al. [85] validated L-valine as a non-invasive urinary biomarker. These data suggest that BCAA dysregulation reflects metabolic stress as well as repair capacity post-injury and is conserved across species.

Purine metabolism emerged in both datasets. In animals, Zeng et al. [58] and Yang et al. [61] identified purine derivatives like AMP, IMP, and uric acid as indicators of energy disruption and oxidative stress. In humans, Bykowski et al. [87] detected 1,3,7-trimethyluric acid and hypoxanthine, linking purine breakdown with poor recovery trajectories. These metabolites may serve as markers of injury burden and mitochondrial dysfunction.

Gut-derived short-chain fatty acid (SCFA) metabolism was another conserved axis. Jing et al. [81] and Kong et al. [88] showed that SCI patients had reduced butyric, acetic, and propionic acids, accompanied by gut dysbiosis. This mirrored findings in animal models (e.g., Wang et al., [63]), where SCFA depletion was associated with neuroinflammation and functional decline. Notably, SCFA supplementation restored motor function in rodents, emphasizing their therapeutic relevance [63].

Lastly, lipid metabolism, including changes in phosphatidylcholines, lysoglycerophospholipids, and arachidonic acid derivatives, was frequently altered across species. Kong et al. [88] in humans and Liang et al. [55] in rats identified these lipids as markers of systemic inflammation and neuroimmune signaling. Zhou et al. [77] further linked exercise-induced lipid shifts to functional recovery, while Lyu et al. [66] mapped lipid species like phosphatidylserine [PS] to lesion regions using mass spectrometry imaging. Collectively, SCI produces a metabolic fingerprint marked by energetic disruption, amino acid and purine dysregulation, gut–brain axis breakdown, and lipid-mediated inflammation. These conserved pathways not only validate cross-species relevance but also represent modifiable targets for biomarker development and therapeutic intervention.

## 5. Potential and Future of Metabolomics in SCI

### 5.1. Potential of Metabolomics in SCI

#### 5.1.1. Injury Profiling and Prognosis

In other clinical fields, metabolomics has already produced validated biomarkers that inform diagnosis, prognosis, and therapeutic decision-making. Maple syrup urine disease, for instance, is detected in newborns by measuring elevated branched-chain amino acids (BCAAs) in dried blood using tandem mass spectrometry [99]. Urinary creatinine–albumin ratios, quantified by mass spectrometry (MS), guide the staging of chronic kidney disease, while serum lactate, also routinely measured by MS, serves as a prognostic marker in sepsis and traumatic shock, reflecting hypoxia and mitochondrial dysfunction [101]. In addition, oncometabolites such as 2-HG are now in clinical use for diagnosing gliomas and assessing mutation status [100].

Building on these precedents, SCI metabolomics studies have begun to reveal injury- and outcome-specific signatures with a similar translational promise. Metabolites such as lactate, acetate, succinate, and β-hydroxybutyrate have been consistently identified in the serum and urine of SCI patients and have been shown to correlate with functional recovery scores (e.g., Spinal Cord Independence Measure) [84,87]. These potential biomarkers reflect core disruptions in mitochondrial function and systemic energy metabolism, common features of acute and subacute SCI.

In animal models, Shi et al. [71] mechanistically linked elevated lactate and glycolytic intermediates to suppressed MCT1-mediated neuronal transport, demonstrating both prognostic and therapeutic relevance. Amino acids such as L-valine and phenylalanine also show prognostic value in both humans and animals [60,73,82,85], echoing their future clinical use in metabolic disorders. Therefore, SCI metabolomics holds the potential to deliver clinically actionable biomarkers that inform early diagnosis, injury stratification, and personalized rehabilitation strategies, provided they undergo rigorous validation, standardization, and integration into clinical practice.

#### 5.1.2. Tailored Therapies

Personalized treatments can be guided by each patient’s metabolic profile, enabling clinicians to target the most disrupted pathways with precision [102]. For example, if metabolomics shows increased reliance on glycolysis or oxidative stress, one strategy is metabolic reprogramming: ketogenic diets have been shown to improve motor recovery and reduce inflammation in SCI patients [83]. Conversely, amino acid or antioxidant supplementation can be adjusted to counteract deficits in individuals. Microbiome–metabolome analysis in BPRA also suggests targeted interventions: Zhang et al. [18] found that avulsion-induced neuropathic pain altered gut *Lactobacillus* and the bile acid–neurotransmitter–amino acid metabolism, contributing to anxiety-like behavior. Supplementing with a *Lactobacillus reuteri*, the dominated probiotic blend reversed these metabolomic changes and relieved anxiety in mice. In human SCI, personalized regimens are under study; for instance, a single-masked, longitudinal, randomized, parallel-controlled study introduced a ketogenic diet during acute SCI care expecting the detection of serum biomarkers linked to improvements in neurological recovery and functional independence [103]. By matching interventions to each patient’s metabolic abnormalities, metabolomics enables precision neurorehabilitation in SCI.

#### 5.1.3. Understanding Secondary Complications

Metabolomics can trace the biochemical roots of post-injury inflammation, oxidative stress, and related complications. After SCI, for example, targeted lipidomics revealed dramatic upregulation of pro-inflammatory arachidonic acid metabolites, notably 5-lipoxygenase and COX-2 products, consistent with intense secondary inflammation [49]. Detection of elevated lactate and glycerol in CSF [104] likewise signals tissue hypoxia and membrane breakdown, linking metabolism to damage. Importantly, multi-omic studies have identified purine and amino acid pathways driving oxidative stress. Integrated transcriptome–metabolome analysis in rats showed that acute SCI caused the profound dysregulation of purine metabolism (e.g., xanthine, inosine), which severely compromised energy balance and increased oxidative stress in the injury microenvironment [58]. Similarly, a mouse SCI model found the accumulation of amino acids [leucine, methionine, phenylalanine, and valine] that correlated with gut dysbiosis and promoted reactive oxygen species generation and neuroinflammation [81]. Metabolomics has illuminated gut–brain axis disruption in SCI, with Jing et al. [81] reporting a 30–40% reduction in key short-chain fatty acids, acetate, propionate, and butyrate due to microbiome shifts in patients; in the same study, SCFA supplementation in mice enhanced neuronal survival, reduced astrogliosis, and improved locomotion, likely by dampening systemic inflammation. Thus, metabolite profiling links immune and metabolic changes after SCI, revealing how molecular byproducts (e.g., oxidized lipids, excitotoxins, and microbial metabolites) underlie secondary injury.

#### 5.1.4. Drug Development and Mechanism-Based Therapeutics

By pinpointing disrupted metabolic pathways, metabolomics can inform the discovery of targeted drugs. Cutting-edge approaches such as dose–response metabolomics and stable isotope-resolved metabolomics are uncovering dysregulated metabolic pathways in diseases, directly informing target identification and validation [105,106]. For example, recent studies have used metabolomics to reveal that restoring phenylalanine levels and modulating glutathione metabolism can identify drug targets such as AKT2 and PDK2 in diabetes and cancer models, thereby guiding precision drug design and repurposing [101,106]. Moreover, artificial intelligence (AI)-driven models, including graph neural networks, are now being applied to predict pharmacokinetic and toxicity profiles with high accuracy, accelerating candidate optimization and reducing late-stage attrition [107]. Additionally, metabolomics is advancing personalized medicine by correlating metabolic signatures (for example, branched-chain amino acids, phosphatidylcholines, and long-chain acylcarnitines in metformin-treated cancers) with treatment efficacy, ensuring therapies align with individual metabolic profiles [108]. Collectively, these innovations are breaking traditional research and development bottlenecks and offering a pathway to overcome Eroom’s Law, delivering targeted, biomarker-driven therapies [101]. In summary, each aberrant metabolic signature uncovered in SCI offers a rational target for drug development, enabling therapies that address the injury’s biochemical mechanisms.

### 5.2. Future Perspectives in SCI Metabolomics

Metabolomics in SCI is at an early stage, facing challenges due to the complexity of metabolic networks and the need for study standardization. Collaborative research integrating various “omics” technologies and expertise from different fields is crucial for understanding SCI and translating findings into clinical practice.

#### 5.2.1. Integrated Systems Biology Approach

A key future direction is to integrate metabolomics with other “omics” (genomics, transcriptomics, and proteomics) to build a holistic view of spinal cord injury pathology and recovery [109]. Single-omics studies capture only one layer of biology, whereas combined multi-omics approaches can reveal complex interactions and compensatory mechanisms that a single modality might miss [58]. Indeed, researchers have begun shifting from isolated analyses toward comprehensive multi-omics strategies to overcome the insufficiencies of any single dataset [110]. Recent SCI studies exemplify this trend: for instance, one clinical trial coupled plasma proteomics with immune cell transcriptomics and then added metabolomic and transcriptomic analysis in an animal model to map the molecular effects of exercise [77]. Such integrative analyses identified key pathways and molecules (e.g., immune and metabolic regulators) that underlie recovery benefits. Similarly, metabolomics combined with transcriptomics can elucidate mechanistic links—as shown by Shi et al. [71], who used metabolite profiling and a lactate sensor to discover a lactate shuttle between endothelial cells and neurons, then re-analyzed single-cell RNA sequencing data to find the corresponding transporter (MCT1) was downregulated after injury [71]. By converging insights from metabolites, gene expression, and proteins, future SCI research can pinpoint biomarker panels and therapeutic targets with far greater precision than metabolomics alone, ultimately providing a more complete “systems biology” understanding of injury and repair.

#### 5.2.2. Technological Advancements

Continued advances in analytical technology are expected to dramatically improve the sensitivity, accuracy, and throughput of SCI metabolomics. Modern mass spectrometry [MS] offers detection limits in the picogram range, orders of magnitude more sensitive than NMR, enabling identification of many more low-abundance metabolites than was previously possible [111]. Emerging MS platforms also provide new capabilities such as spatially resolved metabolite mapping: for example, mass spectrometry imaging allows sensitive label-free detection, quantification, and characterization of metabolites in tissue, adding a spatial context to biochemical changes that conventional MS lacks [112]. In situ mass spectrometry imaging was shown to provide valuable information on the metabolite profiles across various anatomical regions of the spinal cord following SCI, which may support the development of precise treatment strategies for patients [66]. Furthermore, matrix-assisted laser desorption/ionization mass spectrometry imaging (MALDI-MSI), one of the most widely used and recently refined spatial metabolomics techniques, allows for the near single-cell resolution mapping of metabolite distributions across tissue sections, offering powerful insights into localized biochemical processes [113].

On the NMR front, hardware innovations are boosting sensitivity and throughput despite NMR’s lower intrinsic sensitivity. Ultra-high-field magnets (high-frequency spectrometers) and improved cryogenic probes now reveal peaks that were formerly undetectable, and hyperpolarization techniques can amplify NMR signals by >10,000-fold [114], greatly extending the range of observable metabolites. These improvements, combined with better automation and sample processing, are increasing metabolomic workflow throughput without sacrificing data quality. Notably, recent protocols have addressed prior challenges like ion suppression, reproducibility, and spectral complexity in high-throughput MS assays [115]. Both 1H-NMR and MS are already proven tools in SCI biomarker research; for instance, NMR spectroscopy has successfully discovered metabolite biomarkers correlating with recovery in patients [87], and ongoing enhancements in resolution and sensitivity will make future metabolomics studies in SCI even more precise and routinely used in SCI patient care. The net result will be metabolomic analyses that are faster, cover a broader metabolite range, and yield more reliable quantitative data, facilitating their translation into clinical and research settings.

#### 5.2.3. Real-Time Monitoring

Another promising future avenue is the development of real-time metabolomic monitoring for SCI, enabling the dynamic tracking of biochemical changes to inform quick clinical decision-making. Currently, metabolomic analyses are typically performed on discrete samples [e.g., serum, CSF] in a lab setting, but real-time monitoring of endogenous metabolites in vivo could capture the rapid metabolic fluctuations during acute injury or rehabilitation interventions [116]. Such a capability would allow clinicians to detect secondary injury processes or responses to therapy in near real-time and adjust treatments accordingly. Achieving this is challenging due to the issues of sensitivity, speed, and invasiveness [116], but recent technological breakthroughs are starting to make it feasible. For example, new microfluidic and sensor systems are being designed to continuously sample and analyze metabolites from biological fluids, addressing the need for automated, continuous separation and detection of metabolites at the bedside [64,117]. Innovative ambient MS techniques like probe electrospray ionization coupled with tandem MS have already been used in vivo to instantly ionize and identify metabolites from living brain tissue [116], demonstrating the potential for real-time metabolomic readouts in the nervous system. In the context of SCI, researchers have shown that biosensors can directly monitor key injury-related metabolites: Shi et al. used an intracellular lactate sensor to demonstrate lactate shuttling in the cultured cortical cells, providing immediate insights into metabolic support for neurons [69]. In the future, similar biosensors or rapid MS platforms could be deployed to monitor biomarkers (e.g., lactate, glucose, and inflammatory mediators) in SCI patients in intensive care units or during surgery, enabling timely interventions. The ability to obtain metabolic feedback in real time, essentially “metabolic vital signs”, would represent a paradigm shift in SCI management, allowing preemptive adjustments to therapy and more personalized, responsive care [118]. As these technologies mature, real-time metabolomics holds promise for improving outcomes by catching pathological changes early and guiding interventions with unprecedented speed and precision.

#### 5.2.4. Enhancing Rehabilitation (Personalized Nutrition and Exercise)

Metabolomic profiling also holds promise for optimizing rehabilitation interventions, such as tailoring dietary and exercise strategies to individual metabolic needs. Recent studies show that SCI disrupts systemic metabolism and the gut microbiome, leading to deficiencies in beneficial metabolites such as short-chain fatty acids (SCFAs), which play critical roles in neuroinflammation and repair [81,88]. Supplementation with SCFAs or the restoration of SCFA-producing gut bacteria significantly improved motor recovery and reduced inflammation in SCI mouse models [81]. On the exercise front, Zhou et al. [77] found that resistance training induced specific metabolic and immune signatures associated with improved locomotor function in SCI patients. Strikingly, plasma from exercised animals, when transferred to sedentary SCI mice, replicated some of these neuroprotective effects, demonstrating that exercise-induced metabolites mediate functional recovery. Future rehabilitation programs could incorporate metabolomic biomarkers to personalize nutritional support and tailor exercise protocols, thereby enhancing neuroplasticity and repair. This aligns with the growing movement toward precision rehabilitation, where interventions are matched to the individual’s molecular and metabolic state for optimal outcomes.

## 6. Conclusions

Looking ahead, the future of SCI metabolomics lies in scaling up current findings through multi-center, longitudinal studies that validate candidate metabolite panels and correlate them robustly with neurological outcomes. This step is essential for translating pilot data into clinically deployable diagnostic tools. Equally transformative is the integration of machine learning and artificial intelligence, which can manage the high-dimensional complexity of metabolomic datasets to uncover the predictive biosignatures of injury severity and recovery potential. These computational approaches already demonstrate high prognostic accuracy and hold promise for individualized treatment planning, such as patient stratification for clinical trials or tailoring rehabilitation intensity based on metabolic recovery trajectories. Together, these innovations signal a future where AI-enhanced metabolomics drives precision medicine in spinal cord injury.

## Figures and Tables

**Figure 1 ijms-26-06864-f001:**
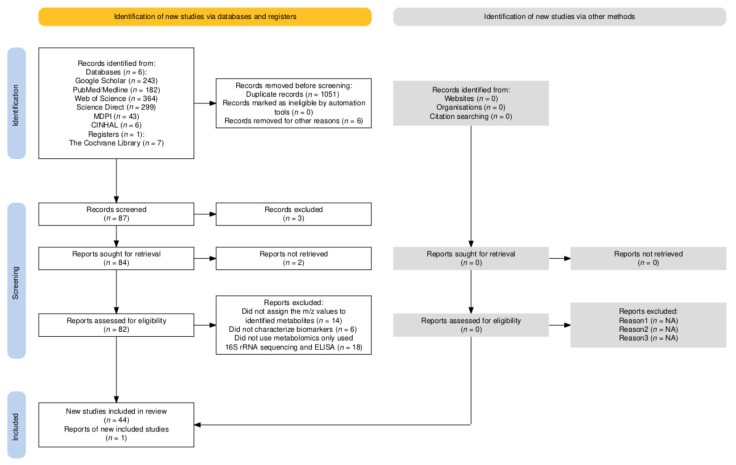
PRISMA study flow diagram. NA—not applicable.

**Table 1 ijms-26-06864-t001:** Overview of recent metabolomics studies in animal SCI models. The table summarizes key metabolites identified, associated biofluids/tissues, metabolomic techniques, and insights into biomarker potential related to spinal cord injury outcomes in rats and mice.

Study	Animal	Tissue Used	Technique	Metabolite	Potential Use as a Biomarker
Pang et al., 2022 [49]	female Wistar rats (*n* = 5/group/time point) at 4 h, 24 h and 48 h post SCI	spinal cord	LC-MS (Liquid Chromatography-Tandem Mass Spectrometry)	Arachidonic metabolites including Prostaglandin E2 (PGE2), Leukotriene B4 (LTB4), Thromboxane B2 (TXB2), HETEs (2-HETE, 8-HETE & 12-HETE), Dihydroxyeicosatrienoic acids (8-DHET, 14-DHETs)	Markers of arachidonic acid metabolism in acute SCI, PGE2 (COX-2 product) and LTB4 were consistently elevated at all three acute time points post-injury and were highlighted as critical inflammatory mediators and potential biomarkers
Domenichiello et al., 2021 [50]	Male Sprague Dawley rats (*n* = 39), *n* = 4 per group for lipidomics,	Dorsal horn of spinal cord, Hind paw skin, sciatic nerve, dorsal root ganglia	LC-MS/MS lipidomics	OXLAMs (Oxidized Linoleic Acid Metabolites) 9-HODE (9-Hydroxy-octadecadienoic acid), oxoODE, 13-HODE, Prostanoids (from Arachidonic Acid) PGE_2_ (Prostaglandin E_2_), TBX2, 5-HETE, 11-HETE, 15-HETE	Oxylipins and OXLAMs associated with nociceptive hypersensitivity and potential pain biomarkers, 5-HETE proposed as possible marker of immune cell infiltration and inflammatory pain.
Rodgers et al., 2022 [51]	Female Long-Evans rats (*n* = 45, 5 per goup),	striatum and spinal cord	LC–MS/MS (Eksigent 425 microLC/SCIEX 5600+ TOF-MS and AB SCIEX 3200 QqQ-MS)	dopamine, L-DOPA, tyrosine pathway metabolites, sphingolipids	propose that dopamine pathway metabolites may serve as markers for opioid responsiveness and targets for overcoming opioid resistance in SCI-related neuropathic pain.
Pukale et al., 2024 [52]	Male Wistar rats (*n* = 20), *n* = 5 per group for metabolomics	spinal cord (syrinx site, rostral & caudal)	LC-MS (liquid chromatography-mass spectrometry) and Mass Spectrometry Imaging (MSI)	betaine, carnitine, alpha-glycerophosphocholine, arginine, creatine, guanidinoacetate, spermidine	Osmolyte biomarkers for syrinx formation and expansion in post-traumatic syringomyelia
Ni et al., 2021 [53]	Rats/*n* = 12 SCI + treatment, *n* = 6 control	spinal cord	LC-MS/MS	Lactic acid, fructose 1,6-bisphosphate, citric acid, glucose 6-phosphate, pyruvic acid	Metabolites proposed as indicators of metabolic dysregulation and inflammation post-SCI
^#^ Zhang et al., 2025 [54]	Male Sprague-Dawley rats; *n* = 8 SCI, *n* = 8 sham (*n* = 36)	serum (paired with gut metagenomics)	Untargeted metabolomics (UPLC-MS/MS for serum);	Pyruvic acid, lactic acid, carnosine, aspartic acid, 3-hydroxyisovaleric acid, isocitric acid, isobutyric acid, ethylmethylacetic acid	Pyruvate and lactic acid are proposed as functional markers of metabolic disturbance and oxidative stress after SCI, and as potential targets for microbiome-based therapeutic interventions
Liang et al., 2024 [55]	Male Sprague-Dawley rats; *n* = 6 per group for metabolomics (*n* = 45)	spinal cord dorsal horn	Untargeted metabolomics (LC-MS/MS) and proteomics	Glycerophospholipids (PC-PC(16:0/18:1), PC(18:0/20:4), PE (18:0/22:6), PS), fatty acyls (Arachidonic acid, Docosahexaenoic acid (DHA), Palmitic acid), steroids (Cholesteryl ester), sphingolipids (Ceramide (Cer))	PEs, PCs, and PA(15:0/20:3-O(5,6)) metabolites serve as markers for neuropathic pain state and therapeutic efficacy, reflecting neuroinflammation and microglial activation status in the spinal cord.
Yuan et al., 2025 [56]	Adult male Sprague Dawley rats, *n* = 6 per group (*n* = 42)	plasma and cerebrospinal fluid (CSF)	LC-MS metabolomics and network pharmacology	dopamine, L-DOPA, normetanephrine, spermine, 5-HTP, phenylpyruvate, N-methyl-L-glutamic acid, carnosine, epinephrine, tryptophan, spermine	5-hydroxytryptophan, normetanephrine, dopamine, tryptophan, and phenylpyruvate are associated with pain modulation and inflammatory response, serving as functional biomarkers of neuropathic pain and therapeutic efficacy of LGZ + SIN through improvements in neurotransmitter balance and inflammation.
Jiang et al., 2023 [57]	Male Sprague-Dawley rats (*n* = 70), metabolomics: *n* = 6/group, 7 days post-SCI	spinal cord and microglial supernatant	GC-MS metabolomics	Docosahexaenoic acid (DHA), ethanolamine	DHA is proposed as a functional metabolite marker for neuroinflammation resolution and repair after SCI, and as a mechanistic biomarker for LbGp efficacy via NF-κB/MAPK signaling in SCI
Zeng et al., 2022 [58]	Adult male Sprague-Dawley rats; *n* = 5 per group for metabolomics (*n* = 28), 3 days post-SCI	spinal cord	Integrated LC-MS/MS metabolomics	purine metabolites: xanthine, inosine, guanidoacetic acid, sphingosine, pantothenic acid, AMP, IMP, GMP, L-glutamine, xanthosine	These purine metabolites are proposed as functional markers of acute SCI microenvironmental disturbance, energy metabolism failure, and targets for neural repair and regeneration strategies
* Liu et al., 2023 [59]	Male Sprague-Dawley rats; *n* = 6 per group (*n* = 42)	bladder muscle, collected at 30 min, 6 h, 12 h, 24 h, 5 d, 2 w post-TSCI	LC-MS untargeted metabolomics	α-ketoglutaric acid, fructose 1,6-bisphosphate, phosphocreatine (energy metabolism); steroid hormones (corticosterone, estrogen, estradiol, 17α-hydroxyprogesterone, etc.); neurotransmitters (acetylcholine, histamine, noradrenaline); ascorbic acid, taurine	α-ketoglutaric acid, fructose 1,6-bisphosphate, and phosphocreatine serve as markers of the three main muscle energy pathways, while steroid hormones and neurotransmitters (acetylcholine, histamine, noradrenaline) reflect stress and neuromuscular signaling changes post-TSCI; ascorbic acid indicates antioxidant response—together, their time-dependent changes act as functional biomarkers of detrusor dysfunction and neurogenic bladder progression, supporting targeted metabolic interventions.
Zhang et al., 2019 [60]	Sprague-Dawley rats; *n* = 10 per group (sham, 12 h, 1 d, 2 d, 7 d post-CEI; total *n* = 50)	serum	UHPLC-Q-TOF-MS (ultra-high-performance liquid chromatography coupled with quadrupole time-of-flight mass spectrometry)	corticosterone, 3b,17a,21-trihydroxypregnenone, L-tryptophan, 5-methoxyindoleacetate, liothyronine, homovanillin, t-leucine, α-ketoisovaleric acid, L-valine, 4-methyl-2-oxopentanoate, xanthosine, docosapentaenoic acid, docosahexaenoic acid, eicosapentaenoic acid, decanoic acid, linoleic acid, palmitic acid, sphingosine, glutathione	Xanthosine, L-tryptophan, glutathione, sphingosine, and DHA proposed as diagnostic and prognostic markers for early-stage cauda equina injury (CEI), providing insight into energy, lipid, amino acid, and nucleotide metabolic changes relevant to injury severity and repair.
Yang et al., 2022 [61]	Female Sprague-Dawley rats; *n* = 6 per group (*n* = 24)	spinal cord, CSF, plasma	Untargeted LC-MS (UHPLC-Q-Exactive Orbitrap)	phosphorylcholine, pyridoxine, guanidoacetic acid, uric acid plus citric acid, creatine, stearic acid, linoleic acid, N-acetylaspartylglutamic acid	Uric acid, phosphorylcholine, pyridoxine, and guanidoacetic acid are proposed as potential metabolite biomarkers for SCI severity assessment and prognosis; These markers may facilitate early diagnosis, severity grading, and translational research in SCI.
Huffman et al., 2023 [62]	Female Sprague Dawley rats; *n* = 15 SCI, *n* = 15 naïve, *n* = 3 sham (for metabolomics: (*n* = 30)	lung tissue, 14 days post-SCI	GC-MS, MALDI-MSI, metabolomics	glutamate, alanine, glycine, palmitate, myo-inositol; N-linked glycans	glutamate, glycine, and N-linked glycosylation are proposed as early biomarkers of SCI-induced acute lung injury (ALI/ARDS). They may serve as sensitive indicators for early detection, monitoring progression, and identifying therapeutic targets for SCI-associated lung injury
Wang et al., 2024 [63]	Male Sprague-Dawley rats; *n* = 5 per group for metabolomics (*n* = 15)	plasma-derived exosomes	Non-targeted UPLC-MS/MS (ACQUITY UPLC-Xevo TQ-S)	butyric acid, isobutyric acid, dihomo-γ-linolenic acid, heptanoic acid, tridecanoic acid	Plasma exosome metabolites associated with depression-like behavior after SCI; potential biomarkers of neuropsychiatric susceptibility
Wu et al., 2023 [64]	Male Sprague-Dawley rats (*n* = 24)	Spinal cord tissue	Untargeted LC-MS metabolomics	Lactic acid, Glutamate, Taurine, GABA, Inosine, Citrate, L-Glutamine, L-Glutamate, N-Acetylneuraminic acid, Phosphorylcholine,	Glutamate is a biomarker of excitotoxicity, Xanthine-oxidative stress, L-Glutamine-inflammation, Creatine-energy metabolism disturbance,
Wu et al., 2020 [65]	Sprague-Dawley rats (*n* = 41)	Lumbar spinal cord segments	LC-MS (untargeted metabolomics	glutamate, lactate, creatine, myo-inositol, and taurine.	Markers of excitotoxicity, mitochondrial dysfunction, energy metabolism disturbance, gliosis, and neuroprotection
Lyu et al., 2025 [66]	Adult female Sprague-Dawley rats (*n* = 12; 6 SCI, 6 sham)	Spinal cord	Ambient air flow-assisted desorption electrospray ionization mass spectrometry imaging (AFADESI-MSI) (Q Exactive Orbitrap)	Lipids (phosphatidylserine [PS], phosphatidylethanolamine [PE], phosphatidylglycerophosphate [PGP], cholesterol ester [CE], ceramide [Cer], phosphatidic acid [PA]), inosine, lactic acid, spermidine	PS, PE, PA, inosine, and spermidine proposed as region-specific markers for inflammation, oxidative stress, and neural repair after SCI; spatial profiling supports precision treatment strategies
Francos-Quijorna et al., 2017 [67]	C57BL/6J mice (*n* = 142)	Whole spinal cord lysate	LC-MS/MS (targeted lipidomics)	PGE2, 14-HDHA, 17-HDHA, 5-HETE, 12-HETE, 15-HETE	Indicators of inflammation onset (PGE2) and delayed resolution (14-HDHA, 17-HDHA); lipid mediators of injury and repair dynamics
Chen et al., 2021 [68]	Male Sprague-Dawley rats; *n* = 8 per group (CCI, sham)	Serum, spinal cord	Untargeted metabolomics (LC-MS/MS)	L-tyrosine, dopamine, 1,4-dihydroxybenzene, anthranilic acid, kynurenic acid, L-histidine, phosphorylcholine, choline, acetyl-DL-leucine, arachidonic acid, indolepropionic acid, glycerophosphocholine, uracil, uric acid, beta-hydroxybutyric acid	beta-hydroxybutyric acid, L-tyrosine, dopamine, anthranilic acid, and kynurenic acid, are proposed as functional serum biomarkers reflecting gut microbiota–metabolite–pain axis disturbances in neuropathic pain.
Shi et al., 2024 [69]	Female C57BL/6J mice 1-day & 28-days postSCI, (*n* = 18, 6 per group)	spinal cord	non-targeted LC-MS/MS (UPLC-HSS T3 C18 column; both positive and negative ion modes)-lipid focused	Neutral lipid droplets, cholesterol crystals, lipid metabolism intermediates	Lipid droplet and cholesterol crystal accumulation may serve as indicators of lipid metabolism dysfunction and chronic inflammation post-SCI; targeting pu.1 shown to reverse these changes thus useful as marker for therapeutic efficacy
Ohnishi et al., 2021 [70]	Mice/*n* = 10 SCI, *n* = 6 sham	Spinal cord	CE-TOF/MS	taurine, glutamate, creatine	Glutamate and taurine proposed as biomarkers of neuronal injury and excitotoxic stress
Shi et al., 2024 [71]	Adult C57BL/6 mice; *n* = 3 per group for metabolomics (sham and 7 days post-SCI) *n* = 18	spinal cord-injury epicentre	LC-MS/MS (targeted energy metabolite profiling)	lactate, glucose-6-phosphate, β-D-fructose-6-phosphate, fructose-6-phosphate, adenosine monophosphate (AMP)	Lactate is proposed as a key marker of metabolic dysfunction and a therapeutic target for restoring neuronal energy supply thereby promoting axon regeneration and functional recovery post-SCI.
* Graham et al., 2019 [72]	C57BL/6 mice; Sham (*n* = 8), SCI + Vehicle (*n* = 9), SCI + SS-31 (*n* = 9) (*n* = 26). 7 days post-SCI	Gastrocnemius muscle	GC-TOF MS (untargeted metabolomics)	glucose-6-phosphate, fructose-6-phosphate, lactic acid, amino acids, creatine, carnitine detected but No significant metabolomic differences between SCI + Vehicle and SCI + SS-31 groups; metabolite variation failed to meet FDR significance,	Although indicators of mitochondrial dysfunction, no single metabolite robustly distinguishing groups; no effect from antioxidant treatment either
^#^ Kang et al., 2023 [73]	Female C57BL/6J mice; *n* = 36 per group (sham, SCI) (*n* = 72). 14 days post-SCI	spinal cord	LC-ESI-MS/MS (widely targeted)	L-leucine, L-methionine, L-phenylalanine, L-isoleucine, L-valine	Branched-chain amino acids strongly correlated with gut microbiota dysbiosis and inflammation via gut-brain axis, proposed as biomarkers of markers of metabolic dysfunction and neuroinflammation, contributing to oxidative stress and inflammatory responses in secondary SCI.
Zhang et al., 2024 [74]	Female C57BL/6J mice (*n* = 3 per group for metabolomics) (*n* = 48), 14 days post-SCI	spinal cord	Targeted LC-ESI-MS/MS with transcriptomic correlation	nicotinamide (NAM), niacinamide, fumaric acid	NAM proposed as a functional metabolite marker and therapeutic candidate for inhibiting fibrotic scar formation and improving outcomes after SCI.
Rong et al. (2024) [75]	Mice (C57BL/6N, *n* = 40)	Feces	Untargeted metabolomics (LC-MS)	Atrolactic acid, 3-(3-hydroxyphenyl) propionic acid, 3-(4-hydroxyphenyl) propionic acid, L-Glutamine, L-Glutamate, L-Methionine, L-Valine, L-Tryptophan, L-Proline, 5-Oxo-D-proline, Taurine, Taurocholate	Identified metabolites showed altered metabolic pathways related to gut-brain communication, inflammation, and neural repair, highlighting their potential as biomarkers for assessing spinal cord injury and treatment efficacy
Scholpa et al., 2024 [76]	Female C57BL/6J mice; *n* = 6 sham, *n* = 8 (7 days post-injury), *n* = 9 (21 days post-injury)	spinal cord	Untargeted metabolomics (LC-MS and GC-MS	phospholipids [PLs]: phosphatidylcholines, phosphatidylserines, phosphatidylethanolamines, phosphatidylglycerols, phosphatidylinositols; lysophospholipids [LPLs]: LPC, LPE, LPS, LPG, LPI; fatty acids [FAs]: arachidonate, DHA, long-chain FAs; carnitine and acylcarnitines; TCA cycle intermediates: citrate, aconitate, fumarate	PL/LPL/FA/carnitine profiles, especially increased LPLs and FAs and decreased carnitine/acylcarnitines, are proposed as biomarkers for injury progression, mitochondrial dysfunction, and the transition from recovery to plateau after SCI.
Zhou et al., 2025 [77]	Mice C57BL/6J mice (*n* = 7–8 per group for plasma metabolomics) (*n* = 24)	plasma	LC-ESI-MS/MS (UPLC, QTRAP SCIEX, targeted and untargeted metabolomics)	Lysoglycerophospholipids/lipid metabolism products (8,15-Dihete, 2-Ethyl-2-hydroxybutyric acid, 3-Hydroxy-4-ethoxybenzoic acid, FFA (16:1), LPE(P-17:0)), S-Allyl-L-cysteine, immunoglobulin heavy chains (IGHG2B, IGHG2C, IGHV5-12, IGHV1-31, IGHV1-82)	Lysoglycerophospholipids and associated immunoglobulin heavy chains are proposed as functional biomarkers and mechanistic mediators of exercise-induced neuroprotection and recovery after SCI
Potter et al., 2023 [78]	Mice/*n* = 8 SCI, *n* = 8 sham, *n* = 8 treated, day 7 and day 28	Skeletal muscle	GC-TOF mass spectroscopy	proline, phenylalanine, lysine, leucine, isoleucine, glucose, fructose, lactate	these metabolites as markers of SCI-induced muscle catabolism, oxidative stress, and altered glucose metabolism
* Graham et al., 2019 [79]	Female C57BL/6 mice; *n* = 5 per group (sham, 7 d post-SCI, 28 d post-SCI) (*n* = 25)	skeletal muscle	Untargeted liquid chromatography-mass spectrometry (LC-MS) or gas chromatography-mass spectrometry (GC-MS).	glucose, pyruvic acid, lactic acid, sorbitol, maltose, maltotriose, oxoproline,	Glucose, lactate, and pyruvate proposed as markers of acute glycolytic dysfunction and glucose uptake impairment in skeletal muscle after SCI
Zhao et al., 2024 [80]	Male C57BL/6 mice (*n* = 48)	Serum, Spinal cord	Untargeted LC MS/MS metabolomics	Alanine (dimethyl), serine, citrate, 5-oxo-L-proline, 2-(2-ethoxyethoxy)-2,4,4-trimethylpentan-3-one, 10,16-heptadecadien-8-ynoic acid, ethanediamide, 2-hexyl-1-decanol	alanine (dimethyl) and serine are biomarkers of altered amino acid metabolism; 2-(2-ethoxyethoxy)-2,4,4-trimethylpentan-3-one and 10,16-heptadecadien-8-ynoic acid are biomarkers of changes in lipid and xenobiotic metabolism,
^#^ Jing et al., 2023 [81]	Female C57BL/6N mice (*n* = 4–6 per group)	Feces, serum	Targeted GC-MS for SCFAs profiling	acetic acid (AA), propionic acid (PA), butyric acid (BA), isobutyric acid, isovaleric acid	AA, PA, and BA are proposed as functional biomarkers of gut-brain axis disruption and as therapeutic markers for inflammation reduction and neurological recovery after SCI.

* Used muscle metabolomics post SCI, # used feaces to correlate with serum metabolomics post SCI.

**Table 2 ijms-26-06864-t002:** Overview of recent metabolomics studies in human SCI models. The table summarizes key metabolites identified, associated biofluids/tissues, metabolomic techniques, and insights into biomarker potential related to spinal cord injury outcomes.

Study	Sample	Tissue Used	Technique	Metabolite	Potential Use as a Biomarker
Singh et al., 2018 [82]	20 ASCI subjects (10 surgical fixation alone, 10 stem cell adjuvant), 10 healthy controls	Serum at baseline and 6 months post-injury	1H NMR spectroscopy (Bruker Avance III 800) untargeted metabolomics	Alanine, acetone, acetate, glucose, formate, glutamine, glycine, threonine, isoleucine, lactate, histidine, phenylalanine, succinate, tyrosine, valine	Glycine, acetone, acetate, lactate, isoleucine, valine, and succinate are potential serum biomarkers for SCI severity and neurological recovery. lactate and glycine inversely correlated with recovery.
Yarar-Fisher et al., 2018 [83]	7 individuals with acute SCI (AIS A-D); randomized to ketogenic diet (*n* = 4) or standard diet (*n* = 3)	Serum	Untargeted LC-MS/MS metabolomics	LysoPC 16:0, fibrinogen alpha and beta subunits	LysoPC 16:0 and fibrinogen may serve as early markers to monitor the efficacy of ketogenic diet interventions and neurological improvement in acute SCI.
Singh et al., 2020 [84]	70 healthy controls 31 ASCI (fixation + stem cell therapy) 34 ASCI (fixation alone)	Urine at baseline, 6 weeks, 3 months, 6 months	1H NMR spectroscopy (Bruker Avance III 800 MHz) untargeted metabolomics	alanine, acetate, β-hydroxybutyrate, choline-containing compounds, creatine, creatine phosphate, creatinine, phenylalanine, propylene glycol, urea.	Alanine, acetate, β-hydroxybutyrate, creatine, phenylalanine, urea are promising non-invasive biomarkers for ASCI severity and neurological recovery.
Bykowski et al., 2021 [85]	6 male SCI patients (4 incomplete, 2 complete)	Paired (6 am & 9 am) Urine samples at 1 month and 6 months post-injury	1H NMR (700 MHz Bruker), quantitative metabolomics	Caffeine, 3-hydroxymandelic acid, L-valine, N-methylhydantoin, dopamine, Sumiki’s acid.	Caffeine, 3-hydroxymandelic acid, L-valine, and N-methylhydantoin are robust, non-invasive urinary biomarkers for SCI recovery and prognosis. Purine and tyrosine metabolism are key pathways.
* Li et al., 2022 [86]	25 adults with SCI (16 Normal Glucose Tolerance, 9 prediabetes/type 2 diabetes)	serum ≥ 3 years post-injury)	LC-MS/MS (Liquid chromatography-tandem mass spectrometry) untargeted metabolomics	Indoxyl sulfate (IS), phenylacetylglutamine, L-5-oxoproline, glutamine	Phenylacetylglutamine and indoxyl sulfate proposed as biomarkers for metabolic and cardiovascular risk in SCI patients. Increased levels reflect dysbiosis and impaired metabolic health in SCI with P/DM.
Bykowski et al., 2023 [87]	7 male SCI patients (5 incomplete, 2 complete)	Paired (6 am & 9 am) serum samples at ~1–3 months and 6 months post-injury	1H NMR (700 MHz Bruker), quantitative metabolomics	1,3,7-trimethyluric acid, 1,9-dimethyluric acid, acetic acid, citric acid, dimethyl sulfone, succinic acid, lactate, D-glucose, D-mannose.	1,3,7-trimethyluric acid, 1,9-dimethyluric acid, and acetic acid are promising serum biomarkers for SCI outcome and recovery (SCIM score). Pathways indicate altered energy and amino acid metabolism post-injury.
* Kong et al., 2023 [88]	11 SCI patients (cervical/thoracic, ASIA A–D, mean duration ~23 months), 10 healthy controls (age/gender matched)	Serum	Untargeted metabolomics (UHPLC-QTOF/MS, Agilent 6550 iFunnel & SCIEX Triple TOF 6600)	Uridine, hypoxanthine, PC(18:2/0:0), kojic acid	Uridine, hypoxanthine, PC(18:2/0:0), and kojic acid are promising serum biomarkers for SCI severity, progression, and therapeutic targeting. gut dysbiosis drives metabolic disturbance.
* Jing et al., 2023 [81]	humans (59 SCI patients, 21 healthy controls)	Feces, serum	Targeted GC-MS for SCFAs profiling	short-chain fatty acids (SCFAs): acetic acid (AA), propionic acid (PA), butyric acid (BA), isobutyric acid, isovaleric acid	AA, PA, and BA are proposed as functional biomarkers of gut-brain axis disruption and as therapeutic markers for inflammation reduction and neurological recovery after SCI.
Zhou et al., 2025 [77]	20 incomplete SCI patients	Plasma of (non-acute, >3 months post-injury, exercise for 4 weeks	LC-ESI-MS/MS (SCIEX QTRAP), untargeted & targeted, MetWare DB	lysoglycerophospholipids (LPE(P-17:0), lipids: 8,15-Dihete, FFA (16:1), PA (18:1(9Z)/18:1(9Z)), S-Allyl-L-cysteine, immunoglobulin heavy chains (IGHG2B, IGHG2C)	Plasma lipid metabolites, especially lysoglycerophospholipids and their correlation with immune proteins, are proposed as biomarkers for rehabilitation response (functional recovery) and neuroprotection in incomplete SCI.
Zhang et al., 2025 [89]	Human (patients with TSCI, *n* = 38; healthy controls, *n* = 21; all male, age 18–60)	Feces	Gas Chromatography-Mass Spectrometry (GC-MS)	Acetic acid, propionic acid, butyric acid, valeric acid, isobutyric acid, isovaleric acid, caproic acid	The altered fecal SCFA profile, specifically reduced butyric and acetic acid and increased isobutyric and isovaleric acid, may serve as biomarkers for neurogenic bowel dysfunction and delayed recovery after TSCI.

* Used muscle metabolomics post SCI.

## Data Availability

No new data were created or analyzed in this study.

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
