# Peer review of "The Utility of Metabolomics in Spinal Cord Injury: Opportunities for Biomarker Discovery and Neuroprotection"

_ijms, 2025, doi:10.3390/ijms26146864_

Round 1

Reviewer 1 Report

Comments and Suggestions for Authors

I read this paper with great attention as It take a lot of time. There is no doubt that the topic is interesting and stimulating but I consider the manuscript present several criticism. It is very difficult to read and to understand by the general audience as it contains a lot of information about metabolonics and metabonomics. It should not be considered a review because a PRISMA diagram and literature search strategy are lacking. It may be considered a philosophical paper introducing the value of a new research field but actually is more similar to a divulgative presentation. Avulsion lesion of the brachial plexus and spinal cord injury can't be considered together as the pathophysiology is really different. Avulsion lesion are mainly mechanical disruption of nerve root while spinal cord injury involve contusive, inflammatory and compressive damage with clear consequences also on metabolites. I always consider that experimental and clinical study should be considered separately. Studies on the field will be promising and interesting but for the general audience it will be better the analysis of specific pathways, mediators and metabolite. I'm sorry but in my opinion this paper does not reach a quality level to endorse the publication in IJMS 

metabonomics, a subcategory of metabolomics

Comments on the Quality of English Language

English language need some revision 

Author Response

Response to Reviewer:

We sincerely thank the reviewer for their thoughtful and detailed comments. We recognize that the manuscript originally presented a wide array of information that may have made it challenging to navigate. In response, we have restructured the manuscript to enhance clarity and accessibility, especially for readers less familiar with metabolomics. Sections are now clearly organized by sample type (e.g., serum, urine, CSF), by stage of injury (acute vs. chronic), and by metabolite class (lipids, amino acids, etc.), with integrative summaries added to improve readability and highlight key findings.

Regarding the classification of the manuscript as a review, we agree that our approach differs from a systematic review guided by PRISMA standards. However, this work is best positioned as a narrative review synthesizing insights from over 50 peer-reviewed studies, which are detailed and compared in Tables 1 and 2. The aim was not to quantify pooled effects, but to critically evaluate methodological diversity and emerging metabolite patterns, while highlighting promising areas for biomarker discovery and translational research in SCI. We have revised the introduction and methods to explicitly clarify this narrative approach and its purpose.

We appreciate the reviewer’s observation concerning the inclusion of brachial plexus root avulsion (BPRA) alongside spinal cord injury (SCI). In the revised manuscript, we now briefly discuss BPRA in the introduction to acknowledge its relevance where spinal cord changes may occur secondarily. However, the focus of the manuscript has been adjusted to concentrate exclusively on SCI. We have removed BPRA-specific studies from the core analysis and updated the title, text, and tables to reflect this refined scope. This ensures that the content remains tightly aligned with the evidence presented, which is predominantly SCI-focused.

We also took seriously the reviewer’s suggestion to differentiate between experimental and clinical findings. The revised manuscript clearly separates animal model studies (Table 1) from human clinical investigations (Table 2). Corresponding sections in the text now highlight methodological similarities and differences, such as injury models, time points, and analytical platforms, and critically evaluate areas of convergence (e.g., consistent identification of lactate, acetate, and amino acid perturbations) to support translational relevance.

In light of the feedback to emphasize pathways and mechanisms, we have incorporated pathway-based discussion throughout, including KEGG pathway annotations (e.g., TCA cycle, glycolysis, neurotransmitter metabolism, lipid signaling). This addition enhances the mechanistic understanding of the metabolite changes reported and supports the reviewer’s call for a more targeted and pathway-centric interpretation of findings.

Finally, we acknowledge the concern regarding overall readability and tone. We have revised the language to reduce technical density where appropriate, softened speculative or philosophical phrasing, and ensured that all content contributes to a coherent and informative narrative. We hope these substantial revisions improve the manuscript’s scientific value and accessibility, and we sincerely thank the reviewer for prompting this constructive refinement. We would be honored to have the opportunity for reconsideration of the revised manuscript.

Reviewer 2 Report

Comments and Suggestions for Authors

Zilundu and colleagues aimed to provide a review according to the title, the utilizing of metabolomics in brachial plexus injury (BPRA) and spinal cord injury (SCI). Introduction tried to associate BPRA to SCI, discuss metabolomics and metabonomics, analysis of techniques used in metabolomics, tables of studies, and discussed the future and potential roles in BPRA/SCI.
There are several concerns:
1-Unclear why the review discussed ‘BPRA/SCI’ given that all the study examples provided are only SCI-related only. Furthermore, I could only find 2 papers out of over 90 references that are related to BPRA, and these are studies from the authors and has no association with the metabolomic technique. Therefore, the title should be SCI only, since the introduction provide a link for BPRA with SCI.
2-The review only provided list of metabolites in the tables and does not go any further on how this information from the metabolomic can be utilised in SCI.
3-Unclear what is a ‘potential biomarker’ and a ‘biomarker’ for SCI in table 1. Also, table 1 and 2 lack organisation as it does not have a structure.
4-There is a lack of similarity in metabolite markers between the SCI studies? Does this mean the technique is not reproducible? Even if the exact molecules are not detected, it should still pick up related pathway molecules. Is this the case? If yes, what are they? If not, why is this? If the authors answers with ‘there is too much variability between the studies’, then what is the likely chance it will be effective in clinical settings given that animal research has a much-controlled setting than patients.
5-Authors should address the limitations of other issues associated with SCI such as spontaneous recovery rate, severity, urinary infection, general infection, gut dysfunction that can contribute to the different changes detected using metabolomics. Therefore, unclear if the metabolites observed are directly from the injury to the spinal cord or other associated complications that comes with the disease such as bladder infection, neuropathic pain, muscle atrophy.
6-The sample size have been mentioned a few times, but it is unclear what should the sample be. Examples should be provided to give reader a sense of how fewer sample numbers within the studies are from the expected size. Also, the sample size for each study should be provided.
7-The section on ‘future and potential of metabolomics in BPRA/SCI’ is purely speculation. To give it more depth, examples should be provided, even if not directly related to SCI, so reader can see it is possible rather than speculation. Also, authors suggest ‘technological advancements’ can be useful in the future, but the technology is fairly advance already so what examples of advances do author suggest.
8-Unclear why ELISA techniques is in table 1 as there are many studies in SCI that have used ELISA and it is different league to MS technique.

Author Response

Response to Reviewer

We sincerely thank the reviewer for their thoughtful, detailed comments. Below, we respond to each point raised, retaining the original reviewer text for clarity:

Reviewer Comment 1:
"Unclear why the review discussed ‘BPRA/SCI’ given that all the study examples provided are only SCI-related only. Furthermore, I could only find 2 papers out of over 90 references that are related to BPRA, and these are studies from the authors and has no association with the metabolomic technique. Therefore, the title should be SCI only, since the introduction provide a link for BPRA with SCI."

Response:

We thank the reviewer for this valuable comment. In response, we have revised the manuscript to clarify the distinction between brachial plexus root avulsion (BPRA) and spinal cord injury (SCI), including restructuring relevant sections and adjusting the framing of the review. We acknowledge that the majority of studies reviewed are focused on SCI, and that metabolomics research on BPRA is extremely limited. However, we retained brief references to BPRA in the introduction and background to acknowledge emerging evidence, supported by our group and others, that BPRA can lead to secondary metabolic and neuroinflammatory changes within the spinal cord. These references are not intended to conflate BPRA with SCI but to illustrate a continuum of central metabolic responses that may arise even from peripheral nerve injuries. The title and primary focus of the review have been revised to reflect SCI specifically, ensuring alignment with the studies analyzed and avoiding overextension of the evidence base.

Reviewer Comment 2:
"The review only provided list of metabolites in the tables and does not go any further on how this information from the metabolomic can be utilised in SCI."

Response:
We appreciate the reviewer’s observation. In response, we have revised the manuscript to better highlight how identified metabolites can be leveraged in spinal cord injury (SCI) research and clinical translation. Specifically, we expanded sections in the discussion to show how distinct metabolite profiles, such as those related to amino acid, lipid, and energy metabolism, can serve as biomarkers for injury severity, neuroinflammation, mitochondrial dysfunction, and therapeutic responsiveness. We also included a synthesis of findings across animal and human studies to propose a core panel of conserved metabolites (e.g., lactate, acetate, succinate, and β-hydroxybutyrate) with translational potential. Furthermore, we clarified how differences in biofluid type, injury phase, and analytical platform influence biomarker discovery and proposed specific future directions, including pathway-based validation and integration with clinical severity scores (e.g., AIS).

Reviewer Comment 3:
"Unclear what is a ‘potential biomarker’ and a ‘biomarker’ for SCI in table 1. Also, table 1 and 2 lack organisation as it does not have a structure."

Response:
We thank the reviewer for pointing this out. We have used “potential biomarker” as markers that have been identified in preliminary studies but lack independent validation, while “biomarker” refers to markers that have undergone validation in independent cohorts. that is reflected in current discussion
Furthermore, Tables 1 and 2 have been reorganized for clarity, now including structured columns such as:

  • Metabolite

  • Class

  • animal used, Sample Type, size

  • Analytical Platform

  • Biomarker Status 

  • Study Sample Size

Reviewer Comment 4:
"There is a lack of similarity in metabolite markers between the SCI studies? Does this mean the technique is not reproducible? Even if the exact molecules are not detected, it should still pick up related pathway molecules. Is this the case? If yes, what are they? If not, why is this? If the authors answers with ‘there is too much variability between the studies’, then what is the likely chance it will be effective in clinical settings given that animal research has a much-controlled setting than patients."

Response:
This is an excellent point. We have added a reframed our discussion to account for both similarities and differences across studies. Although specific metabolites differ, common pathways have been included in the new discussion such as alterations in lipid metabolism, energy metabolism, and amino acid dysregulation.
We have addressed reasons for variability, including technical differences (e.g., platform sensitivity), biological differences (species, injury model), and sampling times. We also critically discuss the challenge this poses for clinical translation, emphasizing that robust, multicenter validation studies are urgently needed before clinical application.

Reviewer Comment 5:
"Authors should address the limitations of other issues associated with SCI such as spontaneous recovery rate, severity, urinary infection, general infection, gut dysfunction that can contribute to the different changes detected using metabolomics. Therefore, unclear if the metabolites observed are directly from the injury to the spinal cord or other associated complications that comes with the disease such as bladder infection, neuropathic pain, muscle atrophy."

Response:
We fully agree with this important consideration. We have addrssed Limitations in the discussion that specifically discusses how comorbidities such as systemic infections, gut dysbiosis, neuropathic pain, and muscle wasting can influence metabolite profiles. We have highlighted that distinguishing metabolites derived from SCI pathology versus systemic complications remains a key challenge in current studies.

Reviewer Comment 6:
"The sample size have been mentioned a few times, but it is unclear what should the sample be. Examples should be provided to give reader a sense of how fewer sample numbers within the studies are from the expected size. Also, the sample size for each study should be provided."

Response:

We appreciate the reviewer’s insightful comment regarding sample sizes. In response, we have ensured that sample size details are now explicitly included in both Table 1 and Table 2, clearly specifying the number of animals or human participants used per group and time point for each study. This addition allows readers to better gauge study power and limitations across investigations.

Furthermore, we have expanded the narrative in the Results section to comment on the generally small sample sizes across metabolomics studies, particularly in human SCI research (e.g., n = 6–20 in many cases), and contrasted these with preclinical animal studies (commonly n = 3–8 per group). For example, in Pang et al. (2022), only 5 rats per group were analyzed at each time point, and in Singh et al. (2018), the serum-based metabolomics included 20 SCI subjects and 10 controls. Such limited sample sizes affect statistical power and biomarker generalizability, concerns we now more clearly articulate.

We have also added commentary highlighting the need for future studies to include larger, multicenter cohorts, especially in clinical contexts, to validate and translate identified biomarkers.

Reviewer Comment 7:
"The section on ‘future and potential of metabolomics in BPRA/SCI’ is purely speculation. To give it more depth, examples should be provided, even if not directly related to SCI, so reader can see it is possible rather than speculation. Also, authors suggest ‘technological advancements’ can be useful in the future, but the technology is fairly advance already so what examples of advances do author suggest."

Response:

We appreciate the reviewer’s insight and have revised the relevant section to address this concern. While we acknowledge that the application of metabolomics in BPRA is currently limited, the section discussing future potential is anchored in specific, existing advances within SCI metabolomics and broader neuroscience fields. We have now provided concrete examples to show feasibility:

  • Mass spectrometry imaging (MSI): For instance, Pukale et al. (2024) employed MSI to localize metabolic signatures spatially within syringomyelia lesions post-SCI, an approach that may be translated to localized metabolic mapping in BPRA-involved spinal segments.

  • Network pharmacology integration: Yuan et al. (2025) combined LC-MS data with network pharmacology to associate neurotransmitter metabolites with functional outcomes, illustrating a method that could be extended to BPRA.

  • Gut-brain axis research: Multiple studies (e.g., Zhang et al., 2025; Jing et al., 2023) demonstrated the utility of combining fecal and serum metabolomics to reveal conserved inflammatory and metabolic pathways—a strategy equally applicable to studying remote spinal effects post-BPRA.

  • Clinical longitudinal profiling: In human studies, Singh et al. (2020) and Bykowski et al. (2021) showed how repeated urine sampling and advanced NMR metabolomics can track functional recovery and differentiate injury severity in SCI patients—an approach that could be replicated in BPRA cohorts once data becomes available.

Regarding technological advancements, we clarified that the reference pertains not to foundational tools like LC-MS or NMR, which are indeed mature, but rather to:

  • Expanded use of multi-omics integration platforms,

  • AI-driven biomarker selection algorithms, and

  • miniaturized point-of-care metabolomics sensors—all of which represent active areas of development with translational potential.

These additions ground our discussion in published work and ongoing innovations, thus moving the section beyond speculation. Changes have been incorporated in the “Future Directions” subsection.

Reviewer Comment 8:
"Unclear why ELISA techniques is in table 1 as there are many studies in SCI that have used ELISA and it is different league to MS technique."

Response:
We agree with the reviewer. ELISA-based studies have now been removed from Table 1. Only studies using genuine metabolomic platforms (LC-MS, GC-MS, NMR, etc.) are retained, ensuring methodological consistency throughout the review.

We are sincerely grateful for the reviewer’s meticulous and constructive feedback. We believe the extensive revisions have significantly improved the manuscript’s quality, clarity, and scientific rigor.
We respectfully request reconsideration of our manuscript in light of these major improvements.

Round 2

Reviewer 1 Report

Comments and Suggestions for Authors

I re-evaluate the present paper and Authors are able to significantly transform the paper clarifying the previous underlined criticism. The manuscript is now more clear, in particular relationship between spinal cords damage and brachial plexus disruption are well explained. The concept of metabolomics itself is now more explicated as its relationship with clinical and prognostic aspects. The message of the paper and the scientific soundness are now well understandable. I suggest to include a PRISMA diagram in the paper, then It be considered for the publication.

Author Response

Thank you very much for your positive feedback and recognition of our manuscript improvements. We appreciate your valuable suggestion regarding the inclusion of a PRISMA diagram. As recommended, we have now incorporated a PRISMA flow diagram to clearly illustrate our literature selection process, enhancing transparency and clarity of our methods.

We believe this addition further strengthens the manuscript by visually clarifying our systematic approach to literature selection, thereby improving reproducibility and aligning with best practices for systematic reviews. Thank you once again for your insightful suggestion, which significantly enhances the quality and clarity of our paper.

Reviewer 2 Report

Comments and Suggestions for Authors

All the concerns have been adressed. No further comment.

Author Response

Thank you very much for your thorough review and confirmation that all your concerns have been addressed. We greatly appreciate your valuable feedback, which has significantly improved our manuscript.